# Paraclinical Changes Occurring in Dairy Cows with Spontaneous Subacute Ruminal Acidosis under Field Conditions

**DOI:** 10.3390/ani12182466

**Published:** 2022-09-18

**Authors:** Doru Morar, Cristina Văduva, Adriana Morar, Mirela Imre, Camelia Tulcan, Kálmán Imre

**Affiliations:** Faculty of Veterinary Medicine, Banat’s University of Agricultural Sciences and Veterinary Medicine, “King Michael I of Romania” from Timișoara, 300645 Timisoara, Romania

**Keywords:** subacute ruminal acidosis, dairy cows, ruminal pH, milk fat-to-protein ratio, hypocalcemia

## Abstract

**Simple Summary:**

Subacute ruminal acidosis is a nutritional disorder that negatively impacts the health, performance, and welfare of dairy cows. This disease evolves subclinically over a long period of time, and finding suitable and easy-to-use biomarkers for early diagnosis is still a challenge for dairy practitioners. This study aimed to investigate the patterns of paraclinical changes and provide valuable data for more accurately identifying subacute ruminal acidosis under field conditions. Changes in certain blood and milk biochemical parameters were identified in this study, which, taken together, could provide a pattern of biochemical changes that would allow the easier identification of subacute ruminal acidosis in dairy cows under field conditions.

**Abstract:**

This study was undertaken to investigate the changes in the blood and milk biochemical parameters found in naturally occurring and long-lasting spontaneous subacute ruminal acidosis (SARA), with the aim of identifying the patterns of paraclinical changes and providing valuable data for more accurately identifying SARA in cows under field conditions. The study was conducted on a dairy herd with a history of the occurrence of SARA-associated clinical signs. Twelve cows, between 20 and 150 days in milk, were randomly selected and subsequently subjected to venous blood, milk, and ruminal fluid collection. The mean pH value of the ruminal fluid was 5.56 ± 0.32, and 58% (7/12) of the tested cows were SARA positive (ruminal pH ≤ 5.5). The albumin, calcium, and phosphorus serum concentration values were significantly lower (*p* < 0.05) in the SARA group than in the group of healthy cows. Serum aspartate aminotransferase (AST) and glutamate dehydrogenase (GLDH) activity were significantly higher in the SARA cows (*p* < 0.05) than in the group of healthy cows. The mean values of milk fat, milk protein content, and milk fat-to-protein ratio were significantly lower (*p* < 0.05) in the tested cows of the SARA group than in the healthy group of cows. In conclusion, the results of the current study indicate that long-term SARA triggered by a high-concentrate diet is associated with clinically significant changes in both the blood composition (hypoalbuminemia, hypocalcemia, and increased serum AST and GLDH activity) and the milk composition (decreased fat and protein percentage and milk fat-to-protein ratio). Altogether, the obtained results provide a more reliable pattern of paraclinical changes and useful insights for detecting SARA in dairy cows under field conditions.

## 1. Introduction

Subacute ruminal acidosis (SARA) is a nutritional disorder that negatively impacts the health, performance, and welfare of dairy cows [1,2]. Generally, it is agreed that SARA is characterized by recurrent daily episodes of ruminal pH decreasing to values below 5.6 for more than 3 h per day [3,4,5]. Feeding diets high in concentrates and low in fibrous forages over a prolonged period are the main cause of SARA [1,2]. No typical clinical signs have been established for SARA, though it is frequently associated with several factors favoring its development and/or clinical manifestations, including a reduction in food intake, the low digestibility of fibrous feeds, diarrhea due to gastrointestinal lesions, liver abscesses, reproductive disorders, and lameness following chronic pododermatitis [6,7]. Moreover, financial losses also result from reduced milk production efficiency, the modification of milk composition, usually due to milk fat and protein depression, and high culling rates due to the complications induced by SARA [5,8].

The clinical signs encountered in cases of SARA are generally nonspecific, making it difficult to establish a diagnosis in farm settings [3,9]. Presently, the most accurate diagnostic method consists of measuring the ruminal fluid pH, sampled by rumenocentesis of the ventral sac [1,10,11]. Unfortunately, this procedure is invasive, and although the measurement of ruminal fluid pH is a critical factor in the disease diagnosis, this method is not feasible under field conditions [3,12]. Therefore, finding alternative, inexpensive, and noninvasive methods to identify cows with low ruminal pH constitutes a challenge for dairy practitioners. In this regard, the use of indirect methods, based on monitoring easily accessible paraclinical parameters, can constitute less expensive tools that are promising for the diagnosis and monitoring of SARA under farm conditions [12,13,14].

Unfortunately, the paraclinical changes that could be helpful in the diagnosis of SARA have been identified, but findings have sometimes been contradictory [15,16,17,18,19]. This study was conducted to investigate the changes in some blood and milk biochemical parameters found in naturally occurring and long-lasting spontaneous SARA, with the aim of identifying the patterns of paraclinical changes and providing valuable data for more accurately identifying SARA in cows under field conditions.

## 2. Materials and Methods

### 2.1. Ethics Statement

This study used the clinical and paraclinical data from a herd of dairy cows in which the ruminal fluid pH and biochemical analyses of blood and milk were performed for diagnostic purposes. The study was carried out with the consent of the owner of the dairy farm, in compliance with the Code of Good Veterinary Practices. The study design was reviewed and approved by the Bioethics Commission of BUASVM Timișoara, Romania.

### 2.2. Animals

The study was carried out on a dairy farm with a history of the occurrence of SARA-associated clinical signs (e.g., intermittent diarrhea with foamy feces and higher amounts of undigested fiber), during a period of at least three months. In addition, the owner reported that the cows had shown decreasing amounts of milk production and progressive weight loss together with overall poor body conditions. The physical exam of the cows in the first 150 days of lactation revealed a body condition score of 2.42 ± 0.31. The dairy farm comprised 62 Holstein lactating cows that were housed in the barn, in tied stalls, and fed at the feed bunk. The cows were fed a diet with separate components comprising approximately 6 kg roughage and 10 kg concentrates per cow, and the dry matter intake was around 14 kg. Fibrous forages included alfalfa hay (4.5 kg), wheat straw (1.5 kg), and concentrate forage composed of maize (82%), wheat bran (10%), sunflower pie (5%), and minerals and vitamins (3%), respectively. The daily amount of the concentrates was divided into two meals, which were administered before the roughage. Before the onset of clinical signs, the daily average milk production for the entire herd was 14.1 kg/cow, with the average lactation milk yield in the herd being 4300 kg. In order to identify the cause of the abovementioned clinical findings, a total of 12 cows, between 20 and 150 days of the lactation period and without mastitis or other inflammatory injuries detectable on physical examination, were randomly selected and subsequently subjected to venous blood, milk, and ruminal fluid collection. From each of the 12 selected cows, a blood sample, a ruminal fluid sample, and two milk samples (one from the morning milking and the other from the evening milking) were collected. The milk sampling was carried out on the day before the collection of the ruminal fluid and blood samples. During the same period, the blood and milk samples from another herd of 12 clinically healthy dairy cows, in the same lactation period (20–150 days), of approximately the same age and the same breed (Holstein) as the tested cows of the SARA group, and with the average daily milk production of 15.4 kg/cow, were collected and analyzed in the same way to compare their results.

### 2.3. Ruminal Fluid Sampling and Analysis

The ruminal fluid was collected from the rumen ventral sac according to the technique described by Nordlund and Garrett [20]. Fluid collection occurred within 3–4 h after delivering the primary concentrate meal because the cows were fed a separate component diet. A stainless-steel needle of 1.6 mm (outer diameter) × 130 mm was introduced into the ventral rumen sac, and approximately 2–3 mL of fluid was drawn into a 5 mL syringe. The pH of the ruminal fluid was measured immediately after sampling using a portable pH meter (InoLab 1130 P, Kladno, Czech Republic), which had previously undergone two-point calibration with pH 4 and 7 buffers. After the ruminal fluid pH measurements, the cows were classified into one of the three following categories: pH < 5.6—SARA positive; pH between 5.6–5.8—SARA marginal; pH > 5.8—SARA negative [20].

### 2.4. Milk Sample Analysis

The milk samples were directly collected from the milking bucket for individual cows, from both the morning and evening milking, immediately after complete milking and after the prior homogenization of the milk. A total of 50 mL of milk was collected at each milking, and it was then mixed with potassium dichromate and stored at 4 °C until analysis. The collected samples were transported in an isothermal box to the Laboratory of Food Hygiene and Technology, the Faculty of Veterinary Medicine, BUASVM Timisoara, where they were immediately processed after their arrival. The determination of the chemical composition of the milk samples, including fat, protein, and lactose concentrations, was conducted via infrared spectrophotometry using a LactoScope FTIR Milk and Dairy Products Analyzer (Delta Instruments, Drachten, The Netherlands).

### 2.5. Blood Sample Analysis

Whole venous blood samples were collected within 2–4 h after the concentrate meals, via coccygeal venipuncture, in a 5 mL vacuum blood collection tube containing gel and clot activator. The vacutainers were transported to the laboratory under refrigerated conditions (<4 °C). Afterward, the serum was separated from the erythrocyte via centrifugation at 3000× *g* for 15 min and then stored at −20 °C until further analysis. Blood serum assays were performed using a fully automated RX Daytona Plus biochemistry analyzer (Randox Laboratories Ltd., Liverpool, UK). The analyzed biochemical parameters, method principles, linearity (*L*), and sensitivity (*S*) were the total protein (Biuret reaction end point; *L*-up to134 g/L, *S*-2.42 g/L); albumin (Bromocresol green; *L*-up to 66.4 g/L; *S*-3.16 g/L); aspartate aminotransferase-AST/GOT (Tris buffer without P5P 37 °C; *L*-up to 927 U/L; *S*-5 U/L); glutamate dehydrogenase GLDH (Triethanolamine buffer 50 mmol, 37 °C; *L*-up to 718 U/L; *S*-3.4 U/L); gamma-glutamyl transferase-GGT (Gamma Glutamyl-3-Carboxy-4-nitroanilide-IFCC, 37 °C; *L*-up to 1559 U/L; *S*-7.5 U/L); alkaline phosphatase-ALP (Diethanolamine buffer DEA 37 °C; *L*-up to 1210 U/L; *S*-15.1 U/L); creatinine (Alkaline picrate no deproteinization; *L*-up to 2505 μmol/L.; *S*-13.5 μmol/L); urea (Urease kinetic; *L*-372 mg/dL, *S*-2.64 mg/dL); calcium (Arsenazo III, *L*-17.3 mg/dL, *S*-0.401 mg/dL); inorganic phosphorus (Phosphomolybdate UV; *L*-11.80 mmol/L, *S*-0.13 mmol/L); D-3Hydroxybutyrate—BHB (enzymatic method, *L*-3.62 mmol/L, *S*-0.05 mmol/L); and nonesterified fatty acids—NEFA (enzymatic method, *L*-6.1 mmol/L, *S*-0.08 mmol/L). The two-point calibration for each determination method included, respectively, a reagent made with Randox Calibration Serum Level 3 (Cat. No. CAL 2351), NEFA CAL Std. (Cat. No. FA 115), Ranbut Standard (cat. No. RB1007/RB1008), and RX series Saline (Cat. No. SA 8396). This assay uses a linear calculation and a daily reagent blank. Quality control was performed with the use of Randox Assayed Multisera, Level 2 (Cat. No. HN 1530) and Level 3 (Cat. No. HE 1532). The daily quality control acceptance requirements were in relation to the multirule quality control (1_3s_, 10_×_).

### 2.6. Statistical Analysis

Statistical analysis was performed using IBM SPSS Advanced Statistics 23.0 (IBM Corp., Armonk, NY, USA), and the results are expressed as means ± standard deviations, including 95% confidence intervals for the mean values. The differences in the mean values of the blood and milk biochemical parameters between the SARA-positive herd and the healthy herd were assessed using one-way ANOVA, and significance was assigned at *p* < 0.05.

## 3. Results

### 3.1. Ruminal Fluid Analysis

In the SARA group, the mean value of the ruminal fluid pH was 5.57 ± 0.34 (minimum 5.12, maximum 6.21). Accordingly, 58% (7/12) of the tested cows were SARA positive (ruminal pH ≤ 5.5), while 25% (3/12) were marginally SARA (pH of 5.6–5.8), and 17% (2/12) were SARA negative.

### 3.2. Milk Sample Analysis

The recorded mean values of the milk fat content and milk protein content were significantly lower (*p* < 0.01) in the tested cows from the SARA group than in those from the healthy group of cows (Table 1).

### 3.3. Blood Sample Analysis

The recorded mean values of the blood biochemical parameters in the SARA and healthy dairy cows are presented in Table 2. The results showed some significant differences (*p* < 0.05) between the blood biochemical parameters of the SARA and healthy groups of cows. The serum concentrations of albumin, calcium, and phosphate were significantly lower (*p* < 0.05) in the SARA group than in the control group. Further, SARA was associated with decreased serum total protein concentration, BHB, and NEFA values, but the differences between SARA and healthy cows were not statistically significant (*p* > 0.05). The serum activity of AST and GLDH were significantly (*p* < 0.05) higher in the SARA cows than in the healthy cows.

## 4. Discussion

In this study, SARA diagnosis in the group of cows under analysis was based on the ruminal fluid pH, with values lower than 5.6 obtained in 58% (7/12) of the tested cows. According to the results of previous studies, if more than 3 (25–30%) of 12 ruminal fluid samples have pH ≤ 5.5, that group of cows can be considered as experiencing SARA [10,20]. The recorded mean ruminal pH of the SARA cows from our study was more severely depressed than that found in the field studies of Gianesella et al. [21] (5.56 vs. 5.8) but was close to the results published by Morgante et al. [22], who showed that the average ruminal pH ranged from 5.59 to 5.77 in the cows from SARA farms.

Low ruminal pH is commonly associated with decreased milk fat concentration, and milk fat depression has, therefore, been frequently used as an indicator of SARA at the farm level [1,3,5]. In our study, the milk fat percentage was significantly decreased in the SARA group of cows, compared with the healthy group of cows. Moreover, 58% of the tested cows in the SARA group had a milk fat percentage of less than 2.5%. In line with the current study, it has been suggested that a milk fat content value of lower than 2.5% in Holstein dairy cows in more than 10% of the herd can be considered an indicator of SARA [23,24].

In agreement with the results obtained in the current study, several studies [8,17,25,26] have shown that a low ruminal pH value may reduce the milk fat content to varying extents, ranging from 7% to 18%. In our study, the milk fat percentage was reduced by 34.2% in the SARA group, compared with the group of healthy cows. In agreement with our results, Enjalbert et al. [19] observed that the milk fat content was reduced by up to 50% in experimentally induced SARA.

In contrast to our study, the published results of other investigations demonstrate that the milk fat content is not influenced by the ruminal pH value [4,15,18,27]. As suggested by Krause and Oetzel [7], these seemingly contradictory results could be related to the length of time of SARA progression. Thus, the results of some studies on short-term-induced SARA showed that the milk fat content was not affected [18,28], while other studies that demonstrated a significant reduction (*p* < 0.05) in the milk fat content had assessed the effect of SARA on the milk fat content over a longer period [29,30]. Moreover, in a study by Chang et al. [30] on dairy cows fed with a high concentrate diet, a significant increase in the milk fat content was observed in the first week in addition to a significant decline in the same compound in the 18th week when compared with dairy cows fed with a low concentrate diet.

Decreases in the milk fat content in SARA are associated with decreases in the acetate-to-propionate ratios and increases in both the insulin levels and the production of trans-octadecenoic acids at the rumen level [5]. A reduction in milk fat synthesis may be related to the low-pH inhibition of the bacteria responsible for the biohydrogenation of fatty acids in the rumen over a long period of time [3,5]. Consequently, the incomplete biohydrogenation of fatty acids increases the rumen concentration of trans-octadecenoic acids, which causes milk fat depression by decreasing the de novo synthesis of milk fatty acids [16,24,31]. Further, Zebeli and Ametaj [32] showed that the rumen LPS-mediated inflammatory responses may be implicated in the lowering of the milk fat content in dairy cows fed with high-grain diets. In a study investigating the effects of liver metabolism on the synthesis of milk fat, it was shown that a long-term high-concentrate feeding diet leads to reduced synthesis and, therefore, decreased concentration of NEFA in the liver, which consequently leads to a decline in milk fat by reducing the levels of the substrate precursor for milk fat synthesis [33].

In our study, the milk protein content was also significantly decreased in the SARA group, compared with the healthy group of cows. In agreement with our results, SARA was found to be associated with decreased milk protein content in the field study of Xu et al. [26]. Further, Li et al. [33] reported that experimentally grain-induced SARA reduced the milk protein content, and Colman et al. [16] observed that the alfalfa–pellet SARA challenge resulted in reduced milk protein concentration. In contrast, some reports have suggested that the milk protein content is increased in SARA cows [5,15], while other studies did not find any influence of SARA on the milk protein concentration [17,29]. However, two recent studies suggested that the long-term feeding of a high-concentrate diet to dairy cows is associated with decreased milk protein content as a result of reduced casein synthesis in the mammary glands caused by the histamine and lipopolysaccharide (LPS) derived from the digestive tract [29,30,34].

The milk fat-to-protein ratio is a common marker for detecting SARA; in general, values lower than 1 may be correlated with SARA [1,32]. In the current study, the milk fat-to-protein ratio was determined as 0.90 ± 0.27, which agrees with the results obtained by Li et al. [35] for experimentally induced SARA. Further, Villot et al. [36] and Nasrollahi et al. [29] found that high-starch diets had a significant impact on lowering the milk fat-to-protein ratio below the threshold value and were indicative of SARA. Nevertheless, the milk fat-to-protein ratio, by itself, is not a sufficient indicator of SARA because it may be impacted by other metabolic or nutritional disturbances [9,36,37].

In the SARA group, we found a significantly lower concentration of serum albumin than in the group of healthy cows. Albumin is synthesized in the liver, and the total body albumin mass is distributed into the blood, interstitial space, and cells [38,39]. Albumin is considered a negative acute phase protein [38], and the low serum albumin concentration may, therefore, have resulted from and reflected the inflammatory condition [39]. Many studies have indicated that SARA induced by a high-grain diet is associated with an inflammatory response [28,40,41]. High-grain diets lead to the depression of the ruminal pH, which can cause the lysis of Gram-negative bacteria and a subsequent increase in the concentration of lipopolysaccharide in the ruminal fluid [42], which can vary between 2- and 14-fold [43,44]. Even if the mechanisms are not fully understood, the translocation of LPS from the ruminal fluid or gastrointestinal tract into the circulation has been reported in some studies [42,43]. LPS translocation to the systemic circulation triggers the release of both proinflammatory cytokines by macrophages in the mesenteric lymph node in addition to Kupffer cells, which may contribute to systemic inflammation [40,43,45]. Inflammatory states increase capillary permeability and the rate of serum albumin escape into the interstitial space [39]. Moreover, albumin is more rapidly degraded in chronic inflammatory states than in healthy states, leading to a decreased serum albumin concentration despite adequate synthesis [39,46].

In the present study, although the serum albumin concentration was decreased in cows with SARA, the serum total protein concentration remained within normal limits, as the serum globulin concentration was increased. This pattern of serum protein concentrations is characteristic of inflammatory conditions, in which hepatic albumin synthesis is decreased, and globulin concentration is mildly increased because the inflammatory cytokines in the liver cause the stimulation of acute phase protein synthesis in addition to the decreased synthesis of albumin and other negative acute phase proteins [38]. Thus, in dairy cows, a strong negative relationship between the positive acute phase protein levels and the serum albumin concentration has been observed [47]. The results of the studies on the effect of SARA on plasma protein concentration are inconsistent. While one study found a significant decrease in the serum protein and globulin concentrations [26], another study found no effect [17], whereas another found increased serum total protein concentration and no effect on serum albumin levels [48].

In this study, the serum calcium concentration was significantly lower (*p* < 0.05) in the SARA group than in the group of healthy cows. Further, the mean value of the serum calcium concentration in the SARA group was below the reference value of the healthy Holstein dairy cows suggested by Cozzi et al. [49]. The most common cause of asymptomatic hypocalcemia in animals is hypoalbuminemia because approximately 40% of the total serum calcium is bound to albumin. However, in this study, only approximately 20–30% of the decrease in the total serum calcium concentration could have been due to decreases in the serum albumin concentration [50]. Therefore, a second cause for hypocalcemia in the SARA group should be considered, because several studies found a decrease in the serum Ca concentration in SARA cows in the absence of hypoalbuminemia [17,44]. Moreover, Danscher et al. [17] found that the ionized calcium concentrations were significantly lower in the blood of the SARA-challenged cows than in the control cows. Although the details of this mechanism are not well-understood, a decreased serum calcium concentration in SARA dairy cows has been associated with the detoxification of the lipopolysaccharides translocated into the bloodstream from the ruminal fluid or the intestinal tract [40,41,44]. Waldron et al. [51] reported that immune activation via intravenous LPS administration led to decreased serum concentrations of serum Ca and P in dairy cows. It was suggested that hypocalcemia might be a protective mechanism against endotoxemia because the lowered concentration of ionized calcium in plasma inhibits LPS aggregation, which creates the necessary conditions for its binding to high-density lipoproteins and, therefore, leads to the neutralization and removal of LPS from the plasma [40,41].

In our study, in addition to the decrease in serum calcium, phosphatemia was significantly lower in the cows from the SARA group than in those from the group of healthy cows. Although phosphatemia is not representative of the total P status of cows, plasma P concentration remains the most commonly used diagnostic measurement available [52]. In dairy cows, the P concentration in the blood is influenced by a large number of factors, including dietary intake, intestinal absorption, the recycling of blood P through saliva, bone resorption, and excretion through feces, urine, and milk [52]. In our study, the low P concentration found in cows with SARA could be the consequence of a combination of factors, such as reduced digestibility of dietary P, the translocation of P from the extracellular to intracellular space, and increased parathormone secretion in response to decreased serum calcium. Several studies have found a reduction in the total tract digestibility of dietary P in lactating dairy cows with grain-induced SARA as a consequence of decreased fiber microbial digestion as well as a poor P release from the fiber in the diet [5,53,54,55]. However, to date, studies have not shown a direct link between the reduced digestibility of dietary P and decreased plasma P concentrations. Instead, SARA has been associated with decreased plasma concentrations of ionized calcium [11], and reduced blood calcium levels have been found to trigger increased PTH release, which promotes the decreases in plasma P by increasing P excretion in saliva and urine [56]. Further, the serum P concentration decreases as a result of P translocation from the extracellular to intracellular space, which occurs due to increased plasma insulin concentration [50]. Grain-induced SARA in dairy cows leads to increased LPS concentrations in the peripheral blood [42], and intramammary or intravenously LPS infusion has been associated with increased circulating insulin levels [57,58].

In our study, the serum activity of hepatocellular leakage enzymes (AST, GLDH) was significantly higher in the SARA cows than in the healthy cows. In ruminants, GLDH and AST are commonly used for the detection of hepatocellular injury, which may result from inflammation, bacterial toxins, toxic chemicals, or metabolic alterations such as hepatocyte lipid accumulation [38]. Previous studies have demonstrated that, on the one hand, grain-induced SARA is associated with increased LPS concentration in the rumen [42,45,54], and, on the other hand, LPS translocation through the rumen and intestinal wall into the portal circulation and even into the systemic circulation is possible [42,43]. LPS in the portal blood is thought to be a major contributor to liver inflammation [59]. Consequently, the increased serum GLDH and AST activity in the SARA cows in this study might have been due to hepatocellular injury, induced by the lipopolysaccharide translocated from the digestive tract into the circulation. These results are similar to those reported in other studies, which showed that low ruminal pH is associated with significant increases in the serum activity of AST [14,26] and GLDH [60] in cows fed a high-grain diet. In contrast, Khiaosa-ard et al. [61] concluded that the serum activity of AST is not associated with decreased ruminal pH in Simmental cows. Although it has been suggested that the serum AST activity might be a possible marker of susceptibility to low ruminal pH in dairy cows [14], AST is not a hepatic-specific enzyme, and serum levels can be influenced by a large number of nonhepatic factors. Therefore, the concomitant measurement of AST with a liver-specific enzyme such as GLDH would increase the specificity in detecting liver injury [38].

## 5. Conclusions

The results of the current investigation revealed that dairy cows fed a high-concentrate diet express clinically significant changes in both the blood compounds (hypoalbuminemia, hypocalcemia, and increased serum GLDH activity) and the milk compounds (decreased fat percentage and milk fat-to-protein ratio). Considered separately, these changes might not be sufficient for the diagnosis of SARA, but their overall consideration might provide a more reliable pattern of paraclinical changes and useful insights for detecting SARA in dairy cows under field conditions.

Taken as a whole, in our study, the blood biochemical profiling of cows with SARA highlighted clinically significant changes in the albumin, calcium, and serum activity of hepatocellular leakage enzymes (AST and GLDH), which were associated with high-magnitude declines in the milk fat percentage and milk fat-to-protein ratio. The main limitation of our study was that its results were obtained from a relatively small number of animals. Additionally, because the results of the current study were obtained from a group of cows with severe and long-lasting SARA, further research is needed to verify whether the biochemical changes found in this study could be used as biomarkers to assist in the early diagnosis of SARA under field conditions.

## Figures and Tables

**Table 1 animals-12-02466-t001:** Results of ANOVA comparing least square mean of milk constituents between SARA and healthy cows.

Milk Constituents (%)	Groups of Cows	Media	Standard Deviation	95% Confidence Interval for Mean		*p*-Value
				Lower Bound	Upper Bound	
Fat	SARA	2.58	0.49	2.27	2.89	<0.01
	Healthy	3.92	0.55	3.57	4.27	
Protein	SARA	2.73	0.19	2.61	2.85	<0.01
	Healthy	3.20	0.37	2.96	3.43	
Lactose	SARA	4.63	0.52	4.30	4.96	0.17
	Healthy	4.93	0.52	4.60	5.26	
Fat-to-protein ratio	SARA	0.93	0.19	0.81	1.05	<0.01
	Healthy	1.20	0.14	1.11	1.29	

SARA: subacute ruminal acidosis.

**Table 2 animals-12-02466-t002:** Results of ANOVA comparing least square mean of serum biochemical parameters between SARA and healthy cows.

Serum Biochemical Parameters	Groups of Cows	Media	Standard Deviation	95% Confidence Interval for Mean		*p*-Value
				Lower Bound	Upper Bound	
Albumin (g/dL)	SARA	2.74	0.36	2.43	3.04	0.001
	Healthy	3.47	0.36	3.21	3.73	
Total protein (g/dL)	SARA	7.64	0.91	6.88	8.41	0.162
	Healthy	8.23	0.77	7.67	8.78	
Calcium (mg/dL)	SARA	7.74	0.19	7.21	8.28	0.001
	Healthy	9.62	0.63	9.16	10.07	
Phosphorus (mg/dL)	SARA	4.95	1.13	3.99	5.90	0.012
	Healthy	6.48	1.11	5.66	7.29	
GGT (U/L)	SARA	26.31	7.34	20.17	32.46	0.39
	Healthy	23.25	7.31	18.02	28.48	
AST (U/L)	SARA	93.36	8.29	57.37	66.84	0.03
	Healthy	61.60	6.26	61.32	126.59	
GLDH (U/L)	SARA	25.89	7.57	20.07	31.71	0.01
	Healthy	12.22	5.97	7.63	16.81	
BHB (mmol/L)	SARA	0.48	0.11	0.40	0.57	0.16
	Healthy	0.60	0.20	0.44	0.75	
NEFA (µmol/L)	SARA	0.21	0.12	0.12	0.31	0.13
	Healthy	0.27	0.14	0.16	0.37	
Creatinine (mg/dL)	SARA	0.98	0,07	0,91	1.04	0.66
	Healthy	1.01	0,19	0.87	1.14	
Urea (mg/dL)	SARA	18.23	5.22	14.91	21.55	0.06
	Healthy	26.32	10.33	19.76	32.89	

SARA: subacute ruminal acidosis; GGT: gamma-glutamyl transferase; AST: aspartate aminotransferase; GLDH: glutamate dehydrogenase; BHB: D-3Hydroxybutyrate; NEFA: nonesterified fatty acids.

## Data Availability

Not applicable.

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
