# Peer review of "Paraclinical Changes Occurring in Dairy Cows with Spontaneous Subacute Ruminal Acidosis under Field Conditions"

_animals, 2022, doi:10.3390/ani12182466_

Round 1

Reviewer 1 Report

Subclinical acidosis is actually a common disease in dairy herds, most often in the period of high daily milk yields of cows. Both the causes and effects of its occurrence are well known. The main reason is the inappropriate structure of the food ration, the excessive supply of unstructured carbohydrates. The effects relate to losses in milk production, negative health consequences, treatment costs and early culling of animals. Thus, the authors undertook research in the still valid and important area.

The layout of the work is correct, the summary is properly prepared, as is the short introduction. However, it is debatable to say that: „…some of the paraclinical changes that could be helpful in the diagnosis of SARA have been identified under experimental or research settings, rather than in field conditions…”. After all, this is the nature of the experiment: different conditions, different nutrition, different lactation period, timing of rumen sampling, etc. The experimental factor determines the potential differences of the analyzed features and indicators.

In line 34: there is "condidions" and should be "conditions".

Comments and questions regarding the methodological part concern the lack of basic information:

SARA is based on information from the breeder-owner.

What was the herd performance level? The breed and the number of animals in the herd?

Animals between 20 and 150 days of lactation? Did this not affect the obtained results?

What was the ration used, what fooders and in what amount? What was its nutritional value? What was the feed intake?

What was the decrease in milk yield in the herd and the condition of cows?

"Conventional" nutrition: concentrate and roughage fed separately, how?

Taking into account the given dose structure - 60% proportion of concentrated feed, it was certainly to be expected that the occurrence of subclinical acidosis would be expected, practically without an analysis.

When were the blood samples collected?

There were 7 cows in the SARA group and 5 (3 + 2) in the healthy group (marginally and negative)?

Line 188: „…herd compared with the healthy herd”, maybe better a group of cows than a herd. This term is used several times in the text.

The analyzed indicators are commonly used in both experimental and production conditions.

The work requires many corrections and additions, especially in the methodological part, in its current version it is not suitable for publication.

Author Response

Reviewer #1

Subclinical acidosis is actually a common disease in dairy herds, most often in the period of high daily milk yields of cows. Both the causes and effects of its occurrence are well known. The main reason is the inappropriate structure of the food ration, the excessive supply of unstructured carbohydrates. The effects relate to losses in milk production, negative health consequences, treatment costs and early culling of animals. Thus, the authors undertook research in the still valid and important area.

Our sincere thanks for taking the time to review this manuscript, and your close attention to detail. Please see below for our responses to your comments:

The layout of the work is correct, the summary is properly prepared, as is the short introduction. However, it is debatable to say that: „…some of the paraclinical changes that could be helpful in the diagnosis of SARA have been identified under experimental or research settings, rather than in field conditions…”. After all, this is the nature of the experiment: different conditions, different nutrition, different lactation period, timing of rumen sampling, etc. The experimental factor determines the potential differences of the analyzed features and indicators.

The authors agree the reviewer comment and, accordingly, the sentences was rephrased into one sentence (see the lines 62-64 of the revised version).

In line 34: there is "condidions" and should be "conditions".

The correction was done.

Comments and questions regarding the methodological part concern the lack of basic information: SARA is based on information from the breeder-owner. What was the herd performance level? The breed and the number of animals in the herd?

According to reviewer suggestion information on the breed, the herd of lactating cows on the farm and milk production/day/cow were included in the text (see the lines 81-82, 88-89 of the revised version).

Animals between 20 and 150 days of lactation? Did this not affect the obtained results?

In a herd of dairy cows, the highest risk of developing SARA is in cows at the early and in the middle of the lactation period.

What was the ration used, what fooders and in what amount? What was its nutritional value? What was the feed intake?

As answers to the raised concerns, information of ration was included in the text (see the lines 84-86, of the revised version).

What was the decrease in milk yield in the herd and the condition of cows?

Milk production decreased by 61.3% (see the lines 88-90 of the revised version).

"Conventional" nutrition: concentrate and roughage fed separately, how?

Cows were fed with concentrate and roughage separately, in the feed trough, for each cow (see the lines 86-88 of the revised version).

Taking into account the given dose structure - 60% proportion of concentrated feed, it was certainly to be expected that the occurrence of subclinical acidosis would be expected, practically without an analysis.

Even if such a ration structure predisposes to the triggering of SARA in cows, a confirmation of the diagnosis through paraclinical analyzes was needed, because the susceptibility of cows to SARA, under the same conditions, is different. In studies conducted by Khiaosa-ard et al., 2018; Gao and Oba, 2014, 2015; Nasrollahi et al., 2017 (included in the reference list of the manuscript) an individual animal variability in susceptibility to SARA in lactating dairy cows has been reported.

When were the blood samples collected?

The blood samples were collected within 2-4 hours after the concentrate meals (see the lines 126-127 of the revised version).

There were 7 cows in the SARA group and 5 (3 + 2) in the healthy group (marginally and negative)?

The number of cows tested from the herd suspected of SARA was 12. A group of cows is defined as having SARA, when more than 30% of the cows from a group of 12 (early or middle lactation) have a ruminal pH below 5.6, according to Garrett et al. (1999); Nordlund and Garrett (1994). As we mention in the manuscript, in our study 58% (7/12) of the cows tested were SARA positive (ruminal pH≤5.5) (see the lines 160-162 of the revised version).

Line 188: „…herd compared with the healthy herd”, maybe better a group of cows than a herd. This term is used several times in the text.

Changed according to the reviewer suggestion.

The work requires many corrections and additions, especially in the methodological part, in its current version it is not suitable for publication.

We hope that all of the raised concerns of the worthy reviewers have been successfully addressed. However, we are ready to further revise/modify the article if still exists some shortcomings.

Thank you again!

Reviewer 2 Report

The paper itself is well written, but it is fundamentally misdirected.

1. Pick a cow that's not in good condition.

2. Check for low pH.

3. Compare various items with healthy cows.

What feed was fed and in what quantities?

There are various reports on the symptoms of SARA, although they are not constant. What is novel about this study ?

Specific comments are as follows,

L25; significantly lower (p < 0.05) in the --- roman type

L27; (p < 0.05) --- roman type

L137-140; roman type

L160; were significantly lower ... in --- roman type

Table 1; Please provide a description of SARA in a footnote.

Check bold font.

Table 2; Please provide a description of SARA, GGT, AST, GLDH, BHB, and NEFA in a footnote.

What are AMLA in Calcium and Phosphorus ?

Check bold font.

in Urea, P-value; 0,06 --- 0.06

L288-; Are P and phosphorus different things?

L303; plasma, ionized calcium --- roman type

L337; insights for for detecting --- insights for detecting

References

Check to see if the journal abbreviation is followed by a period.

Author Response

Reviewer #2

The paper itself is well written, but it is fundamentally misdirected.

  1. Pick a cow that's not in good condition.
  2. Check for low pH.
  3. Compare various items with healthy cows.

Our sincere thanks for taking the time to review this manuscript, and your close attention to detail. Please see below for our responses to your comments:

What feed was fed and in what quantities?

The dairy farm comprised 62 Holstein lactating cows, that were fed a separate components diet comprising approximately 6 kg roughage and 10 kg concentrates per cow. Fibrous forages included alfalfa hay (4,5 kg), wheat straw (1,5 kg), and concentrate forage composed of maize (82%), wheat bran (10%), sunflower pie (5%), minerals and vitamins (3%). The daily amount of concentrates was divided into two meals, which were administered before the roughage (see the lines 82-88 of the revised version).

There are various reports on the symptoms of SARA, although they are not constant. What is novel about this study?

In this study, we did not aim to evaluate the clinical signs of SARA. The aim our study was to identify patterns of paraclinical changes and provide valuable data for identifying SARA in cows under field conditions.

Specific comments are as follows,

L25; significantly lower (p < 0.05) in the --- roman type

The correction was done!

L27; (p < 0.05) --- roman type

The correction was done!

L137-140; roman type

The correction was done!

L160; were significantly lower ... in --- roman type

The correction was done!

Table 1; Please provide a description of SARA in a footnote.

Changed according to the reviewer suggestion.

Check bold font.

The correction was done!

Table 2; Please provide a description of SARA, GGT, AST, GLDH, BHB, and NEFA in a footnote.

Changed according to the reviewer suggestion.

What are AMLA in Calcium and Phosphorus?

The correction was done!

Check bold font.

The correction was done!

in Urea, P-value; 0,06 --- 0.06

The correction was done!

L288-; Are P and phosphorus different things?

The correction was done!

L303; plasma, ionized calcium --- roman type

The correction was done!

L337; insights for for detecting --- insights for detecting

The correction was done!

References

Check to see if the journal abbreviation is followed by a period.

The correction was done!

We hope that all of the raised concerns of the worthy reviewer have been successfully addressed however, we are ready to further revise/modify the article if still exists some shortcomings.

Thank you again!

Reviewer 3 Report

REVIEW

for the journal Animals (ISSN 2076-2615)

Article “Paraclinical Changes Occurring in Dairy Cows with Spontaneous Subacute Ruminal Acidosis under Field Conditions

Manuscript animals-1897262 (Type - Communication)

Authors:  Doru Morar, Cristina Văduva, Adriana Morar, Mirela Imre, Camelia Tulcan, Kálmán Imre

1.This study was undertaken to investigate the changes in blood and milk biochemical parameters found in naturally occurring and long-lasting spontaneous subacute ruminal acidosis (SARA), with the aim of identifying patterns of paraclinical changes and providing valuable data for more accurately identifying SARA in cows under field conditions. The results may help livestock producers to better understand the biological processes and manage this problem related to SARA in cattle more effectively.

2.Lines 74-74. “The study design was reviewed and approved by the Bioethics Commission of BU- 73 ASVM Timișoara, Romania (No……)..”. The study design is not described, so there are uncertainties regarding the duration of the study, the frequency of milk and blood tests, the reasons for sampling and the choice of statistical methods based on the stated objectives.

3.Lines 83-88. “a total of 12 cows, between 20 and 150 days of the lactation period….were randomly selected”. “During the same period, blood and milk samples from another herd of clinically healthy dairy cows were collected and analyzed in the same way to compare the results”.  I have questions about homogeneity and sample size. How did you manage to avoid systematic errors in the experiment? Why the study was not conducted in one herd by selecting a group of healthy cows?

4.As you stated that many different factors can cause ruminal acidosis in dairy cows, and each factor has a different effect on the characteristics studied in the experiments. Did you study the effects of each factor on the results of the experiment?

5.I think that the authors should clearly describe the research design in the methodology section. Authors must provide convincing evidence of sample homogeneity and reduction of systematic error.

6.The statistical analysis section (lines 137-140) is very formal and does not describe what statistical methods and why were chosen to achieve the research objectives, given the research design. I suggest adding this part.

7.Lines 146-147. I recommend moving the analysis of literature sources to the discussion section.

8.Table 1 and Table 2.  “95% Confidence interval for mean”. This indicator was not mentioned in the statistical analysis section of the methodology.

9.Table 1 and Table 2.  In addition, in the description of these tables, the authors repeat the numbers indicated in the tables.

10.I have doubts about the validity of the conclusions presented by the authors, since I doubt that they managed to avoid systematic errors (due to sampling).

11.The article is interesting, but the adjustments mentioned are recommended. Authors should pay attention to the research design, justification of sample selection and their homogeneity.

Sincerely, reviewer.

Author Response

Reviewer #3

1.This study was undertaken to investigate the changes in blood and milk biochemical parameters found in naturally occurring and long-lasting spontaneous subacute ruminal acidosis (SARA), with the aim of identifying patterns of paraclinical changes and providing valuable data for more accurately identifying SARA in cows under field conditions. The results may help livestock producers to better understand the biological processes and manage this problem related to SARA in cattle more effectively.

Our sincere thanks for taking the time to review this manuscript, and your close attention to detail. We highly appreciate your overall positive feed-back regarding the quality of the manuscript! Please see below for our responses to your comments:

2.Lines 74-74. “The study design was reviewed and approved by the Bioethics Commission of BU- 73 ASVM Timișoara, Romania (No……)..”.

The correction was done! (see the lines 74-76).

The study design is not described, so there are uncertainties regarding the duration of the study, the frequency of milk and blood tests, the reasons for sampling and the choice of statistical methods based on the stated objectives.

According to reviewer suggestion the methodology was detailed answering the raised concerns by the reviewer (please see the lines 94-102 of the revised version).

3.Lines 83-88. “a total of 12 cows, between 20 and 150 days of the lactation period….were randomly selected”. “During the same period, blood and milk samples from another herd of clinically healthy dairy cows were collected and analyzed in the same way to compare the results”. I have questions about homogeneity and sample size. How did you manage to avoid systematic errors in the experiment? Why the study was not conducted in one herd by selecting a group of healthy cows?

The recommended procedure for diagnosing SARA on a dairy farm involves measuring the pH of the ruminal fluid of 12 cows at high risk of developing SARA, at early and mid-lactation, respectively. (Kleen et al., 2003; Duffield et al., 2004; Nordlund et al., 1995). In our study, in order to identify the cause of clinical findings, a total of 12 cows, between 20 and 150 days of the lactation period and without mastitis or other inflammatory injury detectable on physical examination, were randomly selected and subsequently subjected to venous blood, milk, and ruminal fluid collection. Because SARA is a condition that evolves over a long period of time sub-clinically, we wanted to compare the results of biochemical analysis of blood and milk from the herd of cows diagnosed with SARA with results obtained by the same methods of analysis from a farm of healthy cows.

4.As you stated that many different factors can cause ruminal acidosis in dairy cows, and each factor has a different effect on the characteristics studied in the experiments. Did you study the effects of each factor on the results of the experiment?

This study was not intended to investigate the factors that lead to the occurrence of SARA in cows. In our study we aimed to identify a pattern of paraclinical changes in blood and milk in herds of cows with SARA under field conditions. In order to achieve this purpose we investigated these changes in a herd of cows diagnosed with SARA compared to healthy cows.

  1. I think that the authors should clearly describe the research design in the methodology section. Authors must provide convincing evidence of sample homogeneity and reduction of systematic error.

According to the reviewer requirement the methodology was detailed (see the lines 82-102 of the revised version).

6.The statistical analysis section (lines 137-140) is very formal and does not describe what statistical methods and why were chosen to achieve the research objectives, given the research design. I suggest adding this part.

Changed according to the reviewer suggestion (see the lines 152-157 of the revised version). The authors statistically interpret the obtained results with the one-way ANOVA test ("analysis of variance"), because this test is the best option to compare the means of two or more independent groups in order to determine whether there is statistical evidence that associated population means are significantly different. In addition, the computed 95% confidence interval values for mean, represents range of values that’s likely to include a population value with a certain degree of confidence. It is expressed as a % whereby a population mean lies between an upper and lower interval.

7.Lines 146-147. I recommend moving the analysis of literature sources to the discussion section.

Changed according to the reviewer suggestion.

8.Table 1 and Table 2. 95% Confidence interval for mean”. This indicator was not mentioned in the statistical analysis section of the methodology.

Changed according to the reviewer suggestion.

9.Table 1 and Table 2. In addition, in the description of these tables, the authors repeat the numbers indicated in the tables.

Changed according to the reviewer suggestion.

  1. I have doubts about the validity of the conclusions presented by the authors, since I doubt that they managed to avoid systematic errors (due to sampling).

Sampling was done according to a validated protocol used for a long time in veterinary medical practice described by Nordlund and Garrett (1994), and Garrett et al. (1999).

11.The article is interesting, but the adjustments mentioned are recommended. Authors should pay attention to the research design, justification of sample selection and their homogeneity.

We hope that all of the raised concerns of the worthy reviewer have been successfully addressed however, we are ready to further revise/modify the article if still exists some shortcomings.

Thank you again!

Round 2

Reviewer 1 Report

The corrections and replies presented can be generally assessed positively. However, in the methodological part, there are still a few issues that have not been fully explained.

The Authors, following the suggestion from Garret et al (1999) about the risk of SARA in the herd, based on the analysis of 12 cows, of which more than 30% have a rumen pH above 5.5 (and not above 5.6), undertook its verification in the field conditions. Maybe it is worth referring to this for the purpose of work and in conclusion?

The decrease in the cows' daily yield was reported, and what about their condition? Cows dry matter intake?

Performance reduction by more than 60% over a three-month period for the entire herd or for a selected group of cows? What was the average lactation milk yield in the herd? What was the cows housing system?

There has been some misunderstanding about the question:  “There were 7 cows in the SARA group and 5 (3 + 2) in the healthy group (marginally and negative)?”, an answer was given: “The number of cows tested from the herd suspected of SARA was 12. A group of cows is defined as having SARA, when more than 30% of the cows from a group of 12 (early or middle lactation) have a ruminal pH below 5.6, according to Garrett et al. (1999); Nordlund and Garrett (1994). As we mention in the manuscript, in our study 58% (7/12) of the cows tested were SARA positive (ruminal pH≤5.5) (see the lines 160-162 of the revised version). However, the question was not about this distribution in the SARA group of cows, but about the numbers of SARA  and healthy cows. The revised version already contains information about a group of 12 healthy cows. In line 97-98: “During the same period, blood and milk samples from another herd of 12 clinically healthy dairy cows…” The values in the group the healthy cows  are a reference to these obtained in the group of sick cows. The content shows that they were animals from a different herd (line 98)? It would be methodically appropriate to compare healthy and diseased cows from one herd. What was the rumen pH value distribution in the group of healthy cows? In Section 3.1 we only have values for SARA cows. It does not appear from the content that rumen samples were also collected from "healthy" cows.

Despite the changes made, I leave the decision to publish the work to the Editorial Office of Animals Journal.

Author Response

Rewiever 1 - round 2

The corrections and replies presented can be generally assessed positively. However, in the methodological part, there are still a few issues that have not been fully explained.

The Authors, following the suggestion from Garret et al (1999) about the risk of SARA in the herd, based on the analysis of 12 cows, of which more than 30% have a rumen pH above 5.5 (and not above 5.6), undertook its verification in the field conditions. Maybe it is worth referring to this for the purpose of work and in conclusion?

The dairy farm comprised 62 Holstein lactating cows, that were housed in the barn, in tied stalls and fed at the feed bunk. The cows were fed a diet with separate components comprising approximately 6 kg roughage, 10 kg concentrates per cow, and the dry matter intake was around 14 kg (see the lines 83-85 of the revised version).

Before the onset of clinical signs, the daily average milk production was 14.1 kg/cow, and over the course of three months milk yield gradually decreased with 61.3% (5.6 kg/cow) (see the lines 89-91 of the revised version).

The decrease in the cows' daily yield was reported, and what about their condition? Cows dry matter intake?

Body condition score for cows in the first 150 days of lactation was 2.42 ±0.31 (see the lines 82-83 of the revised version), and dry matter intake was around 14 kg (see the lines 83-85 of the revised version).

Performance reduction by more than 60% over a three-month period for the entire herd or for a selected group of cows? What was the average lactation milk yield in the herd?

The average lactation milk yield in the herd was 4300 kg (see the line 92 of the revised version).

What was the cows housing system?

As answers to the raised concerns, information of housing system was included in the text (see the lines 83-85 of the revised version).

There has been some misunderstanding about the question: “There were 7 cows in the SARA group and 5 (3 + 2) in the healthy group (marginally and negative)?”, an answer was given: “The number of cows tested from the herd suspected of SARA was 12. A group of cows is defined as having SARA, when more than 30% of the cows from a group of 12 (early or middle lactation) have a ruminal pH below 5.6, according to Garrett et al. (1999); Nordlund and Garrett (1994). As we mention in the manuscript, in our study 58% (7/12) of the cows tested were SARA positive (ruminal pH≤5.5) (see the lines 160-162 of the revised version). However, the question was not about this distribution in the SARA group of cows, but about the numbers of SARA and healthy cows. The revised version already contains information about a group of 12 healthy cows. In line 97-98: “During the same period, blood and milk samples from another herd of 12 clinically healthy dairy cows…” The values in the group the healthy cows are a reference to these obtained in the group of sick cows. The content shows that they were animals from a different herd (line 98)? It would be methodically appropriate to compare healthy and diseased cows from one herd.

In order to assess the metabolic health of a dairy herd, based on metabolic profile tests, it is necessary to compare the results with reference values. Unfortunately, reference values differ depending on the methods of analysis used and can also be influenced by breed, stage of lactation, parity, season of production. For this reason, using the same methods of analysis, we chose to compare the mean values of the biochemical parameters tested in SARA cows with the mean values of the same parameters determined from healthy cows of the same breed (Holstein), stage of lactation, season of production and approximately the same age as the tested cows of the SARA group.

It would certainly have been appropriate to compare healthy and diseased cows from one herd.

Unfortunatelly, the number of lactating cows of the SARA herd was small, and the prevalence of SARA was high. Consequently, for this reason, we chose to compare the results of tests performed in the SARA herd with those from a herd of cows without clinical signs associated with SARA.

What was the rumen pH value distribution in the group of healthy cows? In Section 3.1 we only have values for SARA cows. It does not appear from the content that rumen samples were also collected from "healthy" cows.

Ruminocentesis is an invasive method of diagnosis and it is recommended to be performed only if there is a concern (clinically associated signs) linked to SARA (Oetzel, 2017). For this reason ruminocentesis was not performed in clinically healthy cows.

Despite the changes made, I leave the decision to publish the work to the Editorial Office of Animals Journal.

We hope that all of the raised concerns of the worthy reviewers have been successfully addressed. However, we are ready to further revise/modify the article if still exists some shortcomings.

Thank you again!

Reviewer 2 Report

I understand. It seems to be well corrected, so I have no further comment.

Author Response

Dear reviewer,

Thank you again for your efforts and time reviewing our manuscript!

Best wishes,

The research team 

Reviewer 3 Report

REVIEW

for the journal Animals (ISSN 2076-2615) Article "Paraclinical Changes Occurring in Dairy Cows with Spontaneous Subacute Ruminal Acidosis under Field Conditions" Manuscript animals-1897262 (Type - Communication)

Authors: Doru Morar, Cristina Văduva, Adriana Morar, Mirela Imre, Camelia Tulcan, Kálmán Imre

The authors took my suggestions into account and improved the manuscript, so I have no additional comments.

Reviewer

Author Response

Dear Reviewer,

Thank you again for your efforts and time reviewing our manuscript.

Best wishes,

The research team